# Understanding Interval Change in Chest Radiographs via Temporal Inversion

## Abstract

Recent advances in vision-language pretraining have led to the development of medical-specific models, yet most approaches analyze radiographs in isolation and overlook the essential clinical task of comparing prior and current images to assess patient progression. For chest radiographs, robust modeling of interval change between exams is critical, as clinical reports rely heavily on identifying whether findings have improved, worsened, or remained stable. To address this gap, we present **TILA** (Temporal Inversion-aware Learning and Alignment), a framework that incorporates temporal inversion as a supervisory signal. TILA enforces directional awareness between paired images when their temporal order is reversed, enabling the model to adapt predictions to temporal context while maintaining logical consistency. This inversion-aware alignment is integrated across pretraining, fine-tuning, and inference, offering a complete pipeline for paired CXRs. We also propose a new evaluation protocol to assess model sensitivity, consistency, and robustness under temporal inversion, and introduce the **MS-CXR-T$_{retrieval}$** benchmark for progression-aware retrieval. Experiments on public datasets and real-world hospital records demonstrate that TILA achieves state-of-the-art performance in progression classification, temporal embedding matching, and progression retrieval. These results highlight TILA's potential to advance clinical decision-making through more reliable temporal reasoning and generalizable predictions.

## 1 Introduction

Vision–language pretraining (VLP) models such as CLIP (Radford et al., 2021) and SigLIP (Zhai et al., 2023) have substantially advanced representation learning by aligning visual and textual information, showing strong performance in retrieval and zero-shot classification. Adapting these techniques to medical imaging, however, is challenging due to the complexity of clinical reports and the unique characteristics of radiographs (Zhang et al., 2025; Du et al., 2022).

Medical-specific VLP models have begun to address this by aligning chest X-rays (CXRs) with clinical narratives, enabling disease classification and localization (Zhang et al., 2022; Huang et al., 2021; Boecking et al., 2022; Wang et al., 2022b;a). Yet these approaches typically analyze single CXRs, overlooking the key clinical practice of comparing a patient's current and prior studies to assess interval change (Zhu et al., 2023). Radiologists routinely describe findings as *improved*, *stable*, or *worsened*, emphasizing that temporal context is central to diagnostic reasoning (Hou et al., 2023). Methods that ignore temporal order risk contradictory or unreliable predictions.

To address these temporal relationships, recent studies have developed multi-image encoders specifically designed to analyze disease progression in chest radiographs. For example, BioViL-T (Bannur et al., 2023) incorporates prior images to represent temporal context, and TempA-VLP (Yang & Shen, 2025) employs contrastive objectives to learn dynamic changes across image pairs. Furthermore, methods such as SDPL (Zhu et al., 2024) focus on fine-grained disease progression by disentangling symptom-specific features, while classification-oriented approaches like CheXRelNet (Karwande et al., 2022), CheXRelFormer (Mbakwe et al., 2023), and Med-ST (Yang et al., 2024) leverage graph-based, transformer-based, or mixture-of-expert techniques to capture anatomical and temporal changes. However, despite these advances, most existing approaches rely solely on a single progression classification evaluation, which remains challenging and typically achieves only moderate

accuracy. More importantly, such evaluations do not reveal whether models genuinely understand temporal dynamics or are simply fitting class labels—leaving it unclear if they capture the actual direction of disease progression. Clinically, progression is inherently continuous and *directional*, with labels such as *improved*, *stable*, and *worsened* changing meaningfully when the temporal order of image pairs is reversed. Neglecting this temporal order can produce contradictory or unreliable results, undermining trust in automated progression assessment.

To address these fundamental limitations, we propose TILA, a framework that leverages temporal inversion as a core supervisory signal. TILA systematically integrates temporal inversion throughout both pretraining and fine-tuning, explicitly training models to recognize directional progression between paired CXR images. At inference, TILA enforces directional awareness by combining results from both original and reversed image orders, significantly enhancing the robustness and reliability of clinical decisions. In addition, we introduce a specialized evaluation protocol to measure sensitivity to temporal order inversion and consistency in directional prediction. While radiologists assess progression only in the forward (standard) direction, our *Reversed* and *Combined* settings are designed as analytical tools to validate that predictions in the standard order are reliable and trustworthy. To facilitate clear evaluation, we also develop the MS-CXR-T$_{\text{retrieval}}$ benchmark, specifically designed to assess temporal reasoning capabilities in retrieval scenarios. Our primary contributions can be summarized as follows:

- **Temporal Inversion-aware Learning Framework:** We introduce TILA, a novel approach that incorporates temporal inversion throughout all stages of model training and inference for longitudinal CXR analysis. Specifically, TILA comprises:
  - *Pretraining:* A **Change-aware Sigmoid Loss** to capture directional progression by swapping image pairs during training.
  - *Fine-tuning:* Two loss functions — **Bidirectional Cross-Entropy Loss (BiCE)** and **Temporal Consistency Loss (TCL)** — that enforce consistency and correctness of predictions with respect to temporal order.
  - *Inference:* An **inversion-aware scoring strategy** that fuses predictions from both original and swapped pairs, improving prediction reliability and robustness.
- **Evaluation and Benchmarking:** We develop a dedicated evaluation protocol that leverages temporal inversion to assess a model's sensitivity and consistency with respect to temporal order. In addition, we propose **MS-CXR-T$_{\text{retrieval}}$**, a novel benchmark specifically designed to measure the temporal reasoning abilities of VLP models in clinically relevant retrieval tasks.
- **Comprehensive Experimental Validation:** We demonstrate through extensive experiments that TILA consistently achieves strong performance in diverse evaluation settings, including temporal embedding matching, zero-shot, few-shot learning, and fully supervised progression classification. Results across multiple public benchmarks and real-world clinical datasets confirm TILA's practical effectiveness and strong potential for clinical deployment.

## 2 METHOD

We introduce **TILA**, a framework that leverages **temporal inversion** to model progression in paired CXRs. The key idea is to use inversion during training and inference so the model can recognize directional changes rather than treating progression labels as order-independent.

Our method builds upon BioViL-T(Bannur et al., 2023), a state-of-the-art VLP model designed specifically for temporal CXR analysis. While BioViL-T originally uses a CLIP-style contrastive loss, we adapt it to a more computationally efficient SigLIP loss(Zhai et al., 2023). This adapted model serves as our **baseline** for subsequent temporal inversion mechanisms. As shown in fig. 1, TILA consists of three stages—**pretraining**, **fine-tuning**, and **inference**—each systematically incorporating temporal inversion to enhance temporal reasoning.

### 2.1 PRETRAINING: TEMPORAL INVERSION WITH SIGLIP

**Objective.** The aim of pretraining is to learn robust, temporally-aware representations by leveraging temporal inversion as a contrastive signal. We design a **Change-aware contrastive loss** based on

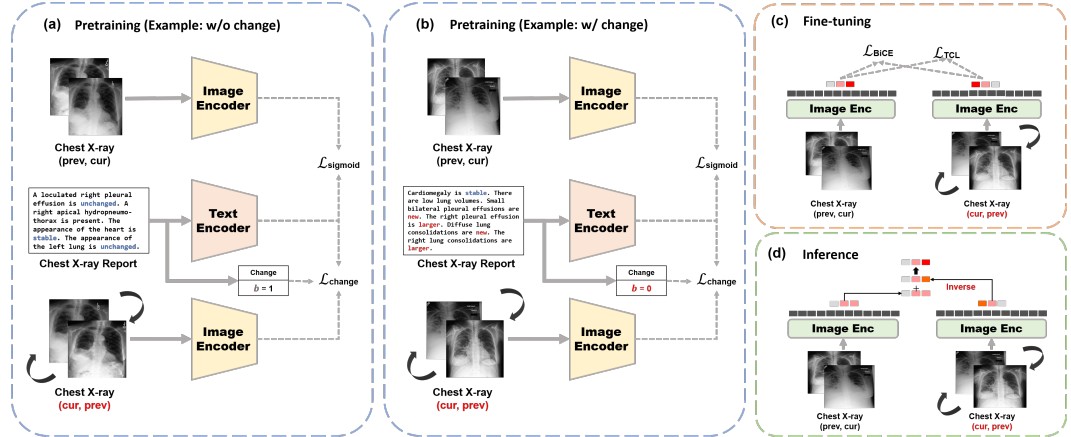

Figure 1: **TILA Framework: Temporal Inversion-aware Learning and Alignment.** The figure illustrates the TILA pipeline, consisting of three stages: pretraining, fine-tuning, and inference. (a, b) **Pretraining:** Paired CXR images (prev, cur) and their reports are encoded, and embeddings are generated for both the original and reversed (cur, prev) image orders. The **Change-aware Sigmoid Loss** aligns pairs for stable reports and distinguishes them for changed reports. (c) **Fine-tuning:** The model uses **Bidirectional Cross-Entropy Loss (BiCE)** and **Temporal Consistency Loss (TCL)** which ensure predictions adapt correctly to swapped image orders and maintain directional consistency. (d) **Inference:** Predictions from both original and reversed pairs are combined using inversion-aware scoring, enhancing robustness by considering both temporal directions.

the sigmoid loss (Zhai et al., 2023), differentiating between stable conditions and genuine temporal changes.

The key insight is that the alignment between image pairs and their associated report should depend on the report's indication of change:

- When the report indicates **no change**, both the original and inverted image pairs should align with the report since the statement remains valid regardless of order.

- However, if the report indicates a **change**, the inverted pair should not align due to the reversed temporal context.

This enables the model to capture progression directionality and to distinguish true progression from stability.

**Image and Text Embedding.** Let $f$ denote the **image encoder** and $g$ denote the **text encoder**. The model takes a pair of longitudinal chest X-rays and a corresponding report as input, producing the following embeddings:

$$v = f(x^{\text{prev}}, x^{\text{cur}}), \quad v^{\text{swap}} = f(x^{\text{cur}}, x^{\text{prev}}), \quad t = g(x^{\text{text}})$$

Here, $v$ represents the embedding from the original order, while $v^{\text{swap}}$ corresponds to the swapped order. The embedding $t$ is derived from the report using the text encoder $g$.

**Temporal Inversion as Supervision.** We automatically label each report as either **change** ($c = 0$) or **no change** ($c = 1$) using the Gemini 2.0 Flash (Team et al., 2023), based on clinically relevant progression keywords (e.g., 'stable', 'improved', 'worsened').

The original Sigmoid loss is:

$$\mathcal{L}_{\text{sigmoid}} = -\frac{1}{|B|} \sum_{i=1}^{|B|} \sum_{j=1}^{|B|} \log \frac{1}{1 + e^{z_{ij}(-\tau \, v_i \cdot t_j + b)}} \tag{1}$$

with

$$z_{ij} = \begin{cases} +1 & \text{if } i = j \\ -1 & \text{if } i \neq j \end{cases}$$

and trainable parameters $\tau$ (logit scale) and $b$ (logit bias).

We further define the change-aware sigmoid loss as:

$$\mathcal{L}_{\text{change}} = -\frac{1}{|B|} \sum_{i=1}^{|B|} \sum_{j=1}^{|B|} \log \frac{1}{1 + e^{z_{ij}^{\text{swap}}(-\tau^{\text{swap}} v_i^{\text{swap}} \cdot t_j + b^{\text{swap}})}} \tag{2}$$

where

$$z_{ij}^{\text{swap}} = \begin{cases} +1 & \text{if } (i = j) \text{ and } (c = 1) \\ -1 & \text{if } (i \neq j) \text{ or } (c = 0) \end{cases}$$

where $\tau^{\text{swap}}$ and $b^{\text{swap}}$ are also trainable parameters.

**Total Loss.** The final pretraining loss combines both losses:

$$\mathcal{L}_{\text{total}} = \mathcal{L}_{\text{sigmoid}} + W \cdot \mathcal{L}_{\text{change}} \tag{3}$$

Here, $W$ is a weighting factor that balances the contributions of both loss components.

By leveraging temporal inversion during pretraining, the model learns to detect changes in a direction-aware manner, aligning model predictions with clinical interpretation of progression.

## 2.2 FINE-TUNING: ENFORCING TEMPORAL INVERSION AWARENESS

**Objective.** Fine-tuning aims to classify progression into three classes—improved, stable, and worsened—while enforcing temporal awareness across image orderings. Importantly, when the image order is swapped, the progression label should also be swapped (e.g., improved becomes worsened in the reversed order).

**Bidirectional Cross-Entropy Loss (BiCE).** To model this label swapping, we define $y$ as the label for the three-class progression classification task, where $y \in \{\text{improved}, \text{stable}, \text{worsened}\}$.

Given a labeled pair $(x^{\text{prev}}, x^{\text{curr}}, y)$, the inversion mapping $\mathcal{I}(y)$ is:

$$\mathcal{I}(y) = \begin{cases} \text{worsened} & \text{if } y = \text{improved} \\ \text{stable} & \text{if } y = \text{stable} \\ \text{improved} & \text{if } y = \text{worsened} \end{cases}$$

The BiCE loss averages the cross-entropy for both the original and reversed pairs:

$$\mathcal{L}_{BiCE} = \frac{1}{2} \left[ \text{CE}(f(x^{\text{prev}}, x^{\text{cur}}), y) + \text{CE}(f(x^{\text{cur}}, x^{\text{prev}}), \mathcal{I}(y)) \right] \tag{4}$$

This ensures the model correctly adapts predictions when the temporal order is reversed, preserving directional consistency.

**Temporal Consistency Loss (TCL).** To further enforce temporal awareness, we require the predicted class probabilities to adjust appropriately under inversion. Let:

- $p_{\text{fwd}}$ denote the predicted probabilities from the original image order (*prior*, *current*).
- $p_{\text{bwd}}$ denote the predicted probabilities from the swapped image order (*current*, *prior*).

We define a probability transformation function $\mathcal{S}$ that swaps the predicted probabilities between improved and worsened while leaving the stable class unchanged:

$$\mathcal{S}(p)[:, 0] = p[:, 2], \quad \mathcal{S}(p)[:, 2] = p[:, 0], \quad \mathcal{S}(p)[:, 1] = p[:, 1]$$

The Temporal Consistency Loss then measures the Mean Squared Error (MSE) between the forward predictions and the transformed reversed predictions:

$$\mathcal{L}_{\text{TCL}} = \frac{1}{N} \sum_{i=1}^{N} \left\| p_{\text{fwd}}^{(i)} - \mathcal{S}(p_{\text{bwd}}^{(i)}) \right\|^2 \tag{5}$$

This loss ensures that the predicted probability distribution appropriately adjusts after inversion, treating progression labels as a continuous scale rather than discrete categories. For example, a high probability assigned to `improved` in the original image order should correspondingly result in a high probability assigned to `worsened` in the reversed order. This fine-grained consistency treats progression as a continuous spectrum and enhances robustness.

**Total Fine-tuning Loss.** The overall fine-tuning objective balances classification and temporal consistency:

$$\mathcal{L}_{\text{total}} = \mathcal{L}_{\text{BiCE}} + \lambda \cdot \mathcal{L}_{\text{TCL}} \tag{6}$$

Here, $\lambda$ is a hyperparameter that controls the trade-off between classification accuracy and consistency across inversion. By applying both BiCE and TCL, the model learns to make order-aware predictions, resulting in more accurate and dependable performance in progression assessment.

### 2.3 INFERENCE: INVERSION-AWARE SCORING

**Objective.** At inference time, we enhance robustness by averaging the predicted probabilities from both the original and reversed image pairs. Specifically, we compute:

$$\text{score} = \frac{1}{2} \left[ p\big(f(x^{\text{prev}}, x^{\text{cur}})\big) + \mathcal{S}\big(p(f(x^{\text{cur}}, x^{\text{prev}}))\big) \right] \tag{7}$$

where $p(\cdot)$ denotes the predicted class probabilities and $\mathcal{S}$ swaps the probabilities for `improved` and `worsened` as defined previously.

By integrating predictions from both temporal directions, this inversion-aware scoring strategy reduces bias and inconsistency, resulting in more reliable progression assessment.

## 3 EXPERIMENTS

We comprehensively evaluate the effectiveness of our proposed **TILA** framework across several key tasks, including retrieval, zero-shot, few-shot learning, and fully supervised progression classification. TILA is compared against state-of-the-art approaches and our own baseline VLP model. Additionally, we introduce **MS-CXR-T$_{\text{retrieval}}$**, a novel benchmark specifically designed to assess temporal alignment for VLP models. Finally, we demonstrate TILA's generalizability through validation on external datasets, including CheXpert (Chambon et al., 2024) and a private cohort from a tertiary hospital.

### 3.1 MODEL IMPLEMENTATION

We use the BioViL-T architecture as the backbone for both image and text encoders, as described in section 2, with the text encoder $g$ initialized from pretrained CXR-BERT (Bannur et al., 2023). Training proceeds in two stages:

- **Pretraining:** The model is first trained for 10 epochs with the original Sigmoid loss. We then introduce the **Change-aware Sigmoid Loss** and continue pretraining for 20 additional epochs, totaling 30 epochs. We set $W = 1$ in the total loss, use a learning rate of $1e-4$, batch size 144, and AdamW optimizer.
- **Fine-tuning:** For progression classification, we fine-tune for 20 epochs using BiCE Loss, then for 30 more epochs with Temporal Consistency Loss (50 epochs total). In few-shot settings, encoder weights are frozen and only linear classifiers are trained; in fully supervised settings, all layers are fine-tuned. We set $\lambda = 50$ for the total loss. Few-shot uses batch size 32 and learning rate $1e-3$, while fully supervised uses batch size 128 and learning rate $1e-5$; AdamW is the optimizer for both.

Full hyperparameter and training details are provided in Appendix A.

### 3.2 DATASET DESCRIPTION

**Training Dataset.** Our primary training data is drawn from MIMIC (Johnson et al., 2016), curated to exclude images without prior counterparts and those overlapping with MS-CXR-T (Bannur et al.,

2023). We utilize Gemini 2.0 Flash (Team et al., 2023) to generate binary change / no change labels for pretraining. For the fine-tuning phase, we use progression labels from Chest ImaGenome (Wu et al., 2021) to perform 3-class temporal classification. For the remaining data, we strictly follow the official data splits to ensure consistency across training, validation, and testing phases.

**MS-CXR-T Dataset.**    MS-CXR-T is a dataset containing paired CXRs from MIMIC, with each pair annotated by radiologists for progression (`improved`, `stable`, `worsened`). It serves as a robust benchmark for temporal progression modeling.

**MS-CXR-T$_{retrieval}$ Benchmark.**    To evaluate temporal retrieval, we introduce **MS-CXR-T$_{retrieval}$**, which reformulates MS-CXR-T as a retrieval challenge. We retrieve original reports matching the images, then modify the progression statements for specific findings using a language model, omitting prior references for unrelated findings. For instance, for a case labeled 'stable' for pneumothorax with the report 'Right pneumothorax is unchanged. Consolidation is worsening.', we generate triplets such as: 'Right pneumothorax is improved. Consolidation is present.', 'Right pneumothorax is unchanged. Consolidation is present.', and 'Right pneumothorax is worsened. Consolidation is present.' This design tests the model's ability to reason about progression in retrieval, allowing comprehensive assessment via all four proposed protocols and including inverse settings.

**External Validation Sets.**    To assess generalizability, we evaluate TILA on external datasets: (i) paired longitudinal images from CheXpert (Chambon et al., 2024) for retrieval and temporal embedding matching, and (ii) a manually annotated set from a tertiary hospital covering four key findings for progression classification validation.

Full dataset and labeling details are provided in Appendix B.

### 3.3 EVALUATION

We assess TILA using standard image-to-text (I2T) and text-to-image (T2I) retrieval metrics (top-$k$ recall) and the TEM score for temporal embedding alignment, where TEM (Bannur et al., 2023) is the F1 of temporal-term overlap ("improved/stable/worsened") between the reference and the retrieved (or generated) report. For classification tasks (zero-shot, few-shot, and fully supervised), we report four complementary evaluation metrics:

- **Standard:** Macro-accuracy on original image pairs, using `improved`, `stable`, and `worsened` labels.
- **Reversed:** Macro-accuracy after reversing image pair order and corresponding progression labels, evaluating the model's ability to recognize directional inversion.
- **Combined:** Combines predictions from both forward and reversed orders as in section 2.3 to generate a final inversion-aware score, enhancing reliability through dual-direction reasoning.
- **Consistency:** Counts a prediction as correct only if the model identifies the correct progression label in both original and reversed orders. This approach ensures that the model consistently outputs accurate decisions from both temporal perspectives, reducing the likelihood of correct predictions by chance.

This comprehensive evaluation protocol measures not only accuracy but also the model's consistency and true understanding of directional temporal changes—key requirements for reliable longitudinal medical imaging analysis.

### 3.4 RESULTS

**Retrieval Performance**    We evaluate retrieval on both the MIMIC and CheXpert datasets, using Recall@1, @5, and @10 for image-to-text (I2T) and text-to-image (T2I) tasks, as well as the TEM score for temporal embedding alignment. As shown in table 1, our baseline model already outperforms the original BioViL-T across nearly all metrics. Notably, the model trained with the TILA pipeline achieves a higher TEM score while maintaining competitive performance in T2I and

Table 1: **Single vs. paired-image retrieval on MIMIC and CheXpert.** Recall@k for I2T/T2I and TEM. Single-image models process only the current image; paired-image models encode (prev, cur). CheXpert results are shown as mean and 95% CI across 10 random subsamples of 3,000 CXRs.

| Model | MIMIC Retrieval | | | | | | | CheXpert Retrieval | | | | | | |
| --- | --- | --- | --- | --- | --- | --- | --- | --- | --- | --- | --- | --- | --- | --- |
| | I2T | | | T2I | | | TEM | I2T | | | T2I | | | TEM |
| | @1 | @5 | @10 | @1 | @5 | @10 | | @1 | @5 | @10 | @1 | @5 | @10 | |
| *Single-image VLPs* | | | | | | | | | | | | | | |
| Ko & Park. (Ko & Park, 2025) | 6.6 | 23.2 | 33.8 | 8.4 | 24.5 | 34.7 | 7.4 | 6.2±0.1 | 16.1±0.2 | 24.7±0.1 | 6.6±0.2 | 15.9±0.2 | 23.0±0.2 | 5.7±0.4 |
| BioViL-T (single image) (Bannur et al., 2023) | 3.1 | 10.7 | 17.9 | 3.7 | 11.3 | 18.4 | 3.5 | 2.4±0.1 | 8.9±0.2 | 14.8±0.2 | 2.6±0.2 | 7.6±0.1 | 13.2±0.1 | 4.2±0.3 |
| BiomedCLIP (Zhang et al., 2025) | 0.4 | 2.0 | 3.1 | 0.6 | 2.3 | 3.2 | 1.6 | 0.2±0.1 | 1.7±0.2 | 2.8±0.3 | 0.6±0.2 | 2.2±0.2 | 3.3±0.3 | 1.9±0.3 |
| *Paired-image VLPs* | | | | | | | | | | | | | | |
| BioViL-T (Bannur et al., 2023) | 5.2 | 17.1 | 25.1 | 5.5 | 16.3 | 24.0 | 12.1 | 3.1±0.1 | 9.8±0.1 | 15.2±0.1 | 2.8±0.1 | 9.2±0.2 | 14.2±0.1 | 13.7±0.2 |
| Baseline | 11.4 | **35.1** | **46.1** | 11.8 | 34.2 | 46.9 | 15.7 | 11.7±0.2 | **31.0±0.1** | **40.6±0.2** | 10.1±0.1 | **26.4±0.2** | **35.8±0.2** | 18.5±0.2 |
| Ours | **12.8** | 34.8 | 45.4 | **12.0** | 33.0 | **47.3** | **17.1** | **12.5±0.2** | 30.2±0.2 | 40.2±0.3 | 10.7±0.2 | 26.3±0.3 | 35.4±0.2 | **20.8±0.1** |

Table 2: Temporal progression classification results on MS-CXR-T and a private dataset, for five findings: consolidation (CON), effusion (PE), pneumonia (PNE), pneumothorax (PTX), and edema (EDE). Columns show four evaluation strategies (Standard, Reversed, Combined, Consistency) as described in Section 3.3.

| Model | Standard | | | | | Reversed | | | | | Combined | | | | | Consistency | | | | |
| --- | --- | --- | --- | --- | --- | --- | --- | --- | --- | --- | --- | --- | --- | --- | --- | --- | --- | --- | --- | --- |
| | CON | PE | PNE | PTX | EDE | CON | PE | PNE | PTX | EDE | CON | PE | PNE | PTX | EDE | CON | PE | PNE | PTX | EDE |
| **Retrieval (MS-CXR-T$_{retrieval}$)** | | | | | | | | | | | | | | | | | | | | |
| BioViL-T (Bannur et al., 2023) | 49.1 | 42.8 | 38.3 | 33.1 | 51.8 | 42.8 | 51.7 | 45.1 | 36.2 | 48.2 | 53.7 | 54.3 | 50.8 | 37.1 | 59.8 | 23.4 | 21.9 | 21.7 | 10.1 | 29.2 |
| Baseline | 48.9 | 58.3 | 55.6 | 34.4 | 53.6 | 51.7 | 53.2 | 61.0 | **38.1** | 50.7 | 55.6 | 60.0 | 64.5 | 39.0 | 58.1 | 32.8 | 35.3 | 40.1 | 14.2 | 41.0 |
| Ours | **55.3** | **58.7** | **58.1** | **40.3** | **58.0** | **55.9** | **59.0** | **61.5** | 37.0 | **56.5** | **58.8** | **62.4** | **65.9** | **46.5** | **62.5** | **39.8** | **39.9** | **43.0** | **22.8** | **44.0** |
| **Zero-shot (MS-CXR-T)** | | | | | | | | | | | | | | | | | | | | |
| BioViL-T (Bannur et al., 2023) | 34.9 | 37.4 | 34.5 | 37.0 | 42.8 | 37.3 | 38.6 | 41.0 | 33.1 | 42.0 | 48.6 | 53.4 | 53.8 | 34.8 | 54.1 | 10.5 | 10.7 | 10.9 | 9.9 | 16.8 |
| Baseline | 42.7 | **50.1** | 42.7 | 29.1 | **55.4** | **47.2** | 43.8 | **49.7** | 35.5 | 51.0 | 46.1 | 50.6 | 51.3 | 33.2 | 58.8 | 21.9 | 23.4 | 27.5 | 7.6 | **39.5** |
| Ours | **46.5** | 46.3 | **51.9** | **38.8** | 52.3 | 46.2 | 46.3 | **49.7** | **38.6** | 52.4 | **56.7** | **58.0** | **59.1** | **42.2** | **60.0** | **27.4** | **24.6** | **30.8** | **33.2** | 36.1 |
| **Few-shot (MS-CXR-T)** | | | | | | | | | | | | | | | | | | | | |
| BioViL-T (Bannur et al., 2023) | 60.3 | 55.6 | 63.7 | 44.3 | 61.6 | 51.1 | 55.3 | 53.4 | 30.8 | 54.9 | 58.2 | 55.2 | 61.1 | 41.3 | 60.6 | 40.3 | 37.7 | 47.7 | 24.2 | 43.4 |
| Baseline | 55.6 | 60.1 | 58.2 | 40.4 | 61.7 | 50.0 | 55.2 | 58.9 | 33.2 | 56.0 | 55.6 | 60.6 | 61.9 | 39.7 | 62.8 | 38.8 | 37.7 | 40.5 | 15.2 | 45.1 |
| Ours | 57.1 | **63.3** | **65.0** | **48.9** | 61.9 | 48.8 | 57.0 | 62.1 | 37.8 | 59.3 | 59.0 | 60.4 | **65.2** | 43.6 | 62.7 | 38.3 | 40.6 | 49.6 | 26.3 | 49.2 |
| Ours + $\mathcal{L}_{BiCE}$ | 60.3 | 60.3 | 63.0 | 43.1 | **64.3** | 59.9 | 59.9 | 63.5 | 43.5 | 58.7 | 63.0 | 61.5 | 64.6 | 43.5 | **63.6** | 51.7 | **46.2** | 51.9 | 24.2 | 55.3 |
| Ours + $\mathcal{L}_{BiCE}$ + $\mathcal{L}_{TCL}$ | **61.2** | 60.0 | 61.3 | 45.5 | 61.2 | **61.0** | **60.2** | 61.7 | **44.3** | **61.8** | **63.4** | 61.5 | 64.4 | **44.2** | 61.7 | **53.2** | 44.4 | **54.9** | **34.1** | **60.1** |
| **Supervised (MS-CXR-T)** | | | | | | | | | | | | | | | | | | | | |
| BioViL-T (Bannur et al., 2023) | 58.1 | 65.7 | 64.7 | 42.9 | 68.9 | 47.5 | 62.3 | 53.5 | 36.9 | 64.9 | 59.2 | 64.2 | 66.3 | 44.5 | 68.4 | 38.2 | 46.6 | 46.8 | 22.7 | 56.3 |
| CheXRelFormer (Mbakwe et al., 2023) | 51.1 | 56.0 | 37.1 | 34.1 | 59.0 | 42.5 | 50.7 | 36.0 | 34.6 | 58.8 | 40.8 | 54.2 | 44.7 | 33.3 | 56.4 | 28.0 | 49.2 | 10.7 | 29.5 | 52.3 |
| SDPL (Zhu et al., 2024) | 45.1 | 49.5 | 63.9 | 29.1 | 62.3 | 41.5 | 20.5 | 42.1 | 19.7 | 24.3 | 43.3 | 37.5 | 53.2 | 25.9 | 53.1 | 39.4 | 18.3 | 16.1 | 14.7 | 15.2 |
| CNN-TF (Bannur et al., 2023) | 46.6 | 55.2 | 41.3 | 35.2 | 61.2 | 37.8 | 39.4 | 32.1 | 22.7 | 47.1 | 48.3 | 57.1 | 44.0 | 37.9 | 60.1 | 27.4 | 25.2 | 27.6 | 14.0 | 34.1 |
| CNN-TF + $\mathcal{L}_{BiCE}$ | 53.2 | 58.2 | 44.1 | 38.4 | 61.0 | 45.6 | 52.3 | 40.0 | 34.5 | 62.1 | 55.4 | 59.5 | 50.8 | 40.1 | 62.1 | 37.4 | 39.4 | 42.7 | 28.9 | 49.2 |
| CNN-TF + $\mathcal{L}_{BiCE}$ + $\mathcal{L}_{TCL}$ | 48.6 | 55.8 | 42.9 | 37.5 | 60.7 | 43.6 | 49.7 | 38.5 | 37.6 | 61.7 | 52.2 | 57.8 | 49.2 | 39.7 | 62.4 | 36.1 | 40.5 | 44.2 | 26.6 | 52.8 |
| Baseline | 61.3 | 67.5 | 64.2 | 43.0 | 69.3 | 52.6 | 61.4 | 56.1 | 32.6 | 63.8 | 60.4 | 65.6 | 63.0 | 42.2 | 67.0 | 41.3 | 40.7 | 45.2 | 16.6 | 53.6 |
| Baseline + $\mathcal{L}_{BiCE}$ | 60.5 | 66.4 | 63.3 | 47.2 | 68.2 | 56.3 | 64.6 | 58.2 | 39.9 | 68.3 | 61.4 | 66.6 | 64.1 | 44.2 | 70.0 | 46.3 | 48.1 | 49.1 | 28.9 | 62.1 |
| Baseline + $\mathcal{L}_{BiCE}$ + $\mathcal{L}_{TCL}$ | 62.3 | 67.2 | 65.5 | 45.5 | 68.7 | 60.9 | 65.7 | 62.5 | 39.5 | 68.9 | 62.1 | 66.1 | 66.2 | 43.6 | 68.7 | 49.8 | 55.6 | 55.0 | 32.2 | 64.4 |
| Ours | **65.2** | **70.6** | 70.2 | 47.3 | 69.7 | 56.9 | 65.1 | 60.9 | 42.1 | 66.7 | **65.5** | **67.0** | 66.9 | 47.9 | 69.6 | 46.3 | 48.1 | 47.7 | 27.5 | 57.6 |
| Ours + $\mathcal{L}_{BiCE}$ | 63.2 | 67.4 | **70.4** | **54.4** | 70.7 | 60.0 | 64.5 | 65.9 | 51.0 | 68.4 | 63.6 | 66.0 | 66.6 | **51.2** | **69.7** | 52.7 | 52.2 | 54.0 | 41.2 | 65.2 |
| Ours + $\mathcal{L}_{BiCE}$ + $\mathcal{L}_{TCL}$ | 63.3 | 66.7 | 67.1 | 52.3 | **70.9** | **64.5** | **65.8** | **66.6** | **51.2** | **70.3** | 63.7 | 66.9 | **67.6** | 51.0 | 68.8 | **60.2** | **58.3** | **57.8** | **43.1** | **67.4** |
| **Supervised (Private)** | | | | | | | | | | | | | | | | | | | | |
| CNN-TF (Bannur et al., 2023) | 41.0 | 47.5 | 35.9 | 31.2 | - | 33.8 | 40.4 | 30.6 | 31.1 | - | 43.7 | 49.0 | 36.1 | 32.3 | - | 21.9 | 22.6 | 13.8 | 12.9 | - |
| BioViL-T (Bannur et al., 2023) | 55.0 | 57.1 | 60.3 | 49.7 | - | 44.8 | 48.1 | 50.2 | 53.4 | - | 57.6 | 56.7 | 54.5 | 53.7 | - | 29.8 | 34.0 | 33.3 | 25.1 | - |
| CheXRelFormer (Mbakwe et al., 2023) | 47.9 | 48.4 | 41.3 | 38.9 | - | 41.8 | 49.2 | 36.2 | 33.0 | - | 49.7 | 48.3 | 41.7 | 34.6 | - | 26.5 | 43.4 | 19.0 | 26.2 | - |
| Baseline | 53.9 | 59.3 | 51.3 | 56.6 | - | 45.6 | 53.1 | 48.8 | 48.1 | - | 56.0 | 59.3 | 54.9 | 52.0 | - | 30.4 | 35.6 | 34.7 | 23.7 | - |
| Ours | 58.2 | 60.2 | 52.6 | 57.7 | - | 46.8 | 54.5 | 48.2 | 52.8 | - | 61.8 | 59.4 | 55.1 | 60.1 | - | 33.8 | 36.6 | 34.0 | 37.1 | - |
| Ours + $\mathcal{L}_{BiCE}$ | 64.3 | 62.3 | 56.5 | 58.6 | - | 57.2 | 56.9 | 53.1 | **62.8** | - | **66.9** | 60.6 | **59.1** | 62.2 | - | 51.2 | 47.6 | 43.4 | 46.8 | - |
| Ours + $\mathcal{L}_{BiCE}$ + $\mathcal{L}_{TCL}$ | **67.9** | **70.2** | **63.0** | **63.3** | - | **61.5** | **65.7** | **55.2** | 62.3 | - | 64.7 | **66.8** | 58.3 | 62.2 | - | **58.4** | **55.2** | **53.4** | **51.4** | - |

I2T retrieval, indicating that change-aware supervision improves sensitivity to progression-related phrases.

In addition, we extend our comparison to single-image vision–language models. As seen in the results, BioViL-T already demonstrates relatively lower retrieval scores but substantially higher TEM compared to single-image models (e.g., TEM 12.1 vs. ≤ 7.4). This highlights that encoding paired images is essential for capturing temporal semantics: paired embeddings not only elevate the TEM score but also improve retrieval robustness. Building on this, our TILA model further increases both retrieval accuracy and TEM, underscoring that temporal inversion is a key ingredient for reliable progression-aware retrieval.

**Temporal Retrieval Performance** As shown in table 2, we evaluate temporal retrieval performance using the newly created **MS-CXR-T$_{retrieval}$** benchmark. Our model, trained with the Change-aware Sigmoid loss, consistently outperforms both the baseline and the original BioViL-T, particularly in detecting changes related to consolidation and pneumothorax. Notably, our approach also achieves

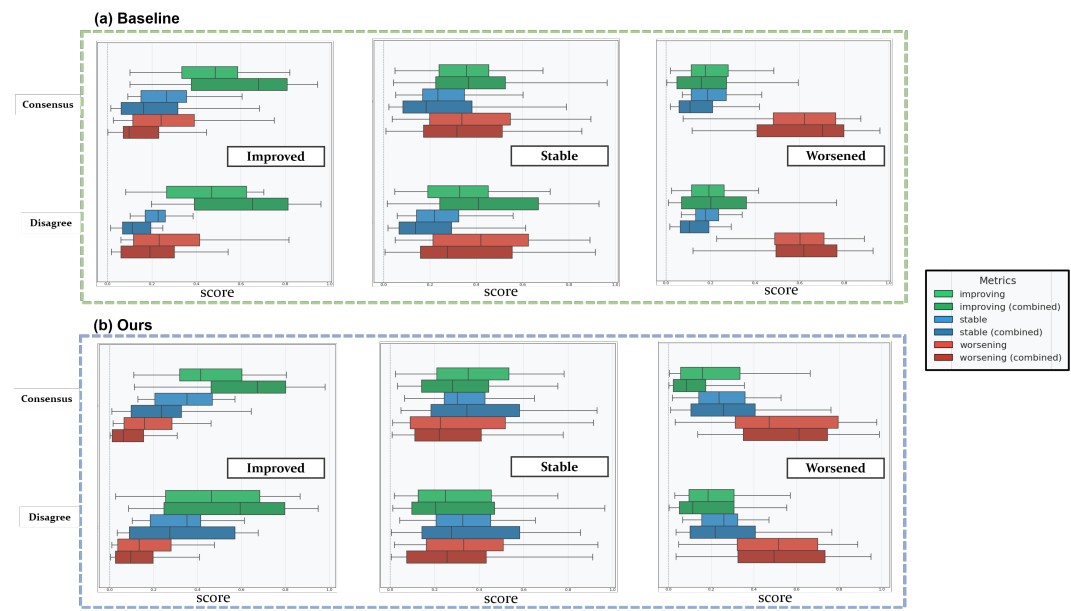

Figure 2: **Score Distribution Analysis for Pleural Effusion (MS-CXR-T).** Distribution of prediction scores (scaled by 10) for each pleural effusion progression label, comparing baseline and TILA models in the zero-shot setting. Each boxplot separates cases by progression label (`improved`, `stable`, `worsened`) and label quality (`consensus` vs. `disagreement`), shown for both standard and inversion-aware (combined) scoring. Key trends illustrated: i) Changes in scores for each label (`improved`, `stable`, `worsened`). ii) Impact of applying inversion-aware scoring (combined). iii) Differences in scores based on label quality (`consensus` vs. `disagreement`).

high consistency scores, confirming that performance gains reflect genuine temporal reasoning rather than random guessing. Additionally, we observe that applying the combined scoring inference method—integrating predictions from both temporal directions—improves accuracy for all models. This likely reflects the method's ability to resolve label ambiguities and reduce uncertainty by enforcing directional consistency across predictions.

**Zero-Shot Performance** We assess zero-shot classification on MS-CXR-T across all evaluation protocols: standard, reversed, combined, and consistency. As detailed in table 2, models pretrained with TILA consistently outperform baselines, with especially pronounced improvements for challenging findings such as pneumothorax. These results highlight the impact of temporal inversion during pretraining, which enhances the model's ability to recognize clinical progression patterns without any task-specific fine-tuning.

**Few-Shot and Fully-Supervised Performance** In both few-shot and fully supervised settings, we conduct ablation studies to evaluate the contributions of each TILA fine-tuning component. As shown in table 2, incorporating the BiCE loss markedly improves classification accuracy over standard objectives. Further addition of the TCL consistently boosts directional robustness, resulting in predictions that are not only accurate but also temporally consistent. Notably, applying TILA techniques across different backbone architectures yields consistent gains, underscoring the broad applicability and effectiveness of our inversion-aware approach.

**External Validation** On an external validation set from a private tertiary hospital (table 2), the TILA framework consistently outperforms baseline and competing methods. These findings demonstrate enhanced reliability and support the potential for improved generalization, suggesting that TILA is well suited for deployment in diverse clinical settings.

**Score Analysis** To investigate how TILA influences the model's scoring behavior, we perform a detailed analysis focusing on effusion predictions from MS-CXR-T. The label quality is categorized

as `multiple experts`, `one-expert`, or `disagreement`. A `multiple experts` label indicates consensus among radiologists, whereas `disagreement` reflects variability, where one radiologist may label a case as `worsened` or `improved` while another labels it as `stable`. Due to the ordinal nature of progression, a `worsened` label with a `disagreement` quality is likely to be closer to `stable`.

As shown in fig. 2, inversion-aware scoring (section 2.3) consistently shifts model predictions toward the correct label, increasing score separation and moving incorrect scores further away. This effect is most pronounced in consensus cases, suggesting that integrating both temporal directions reduces ambiguity and sharpens calibration. Notably, for disagreement cases labeled as `improved` or `worsened`, TILA exhibits smaller score shifts, reflecting more cautious behavior in uncertain settings. This aligns with radiologists' tendencies to avoid overconfident predictions when faced with ambiguous evidence.

Importantly, TILA demonstrates superior ability to distinguish `stable` from `improved` or `worsened` pairs. The baseline model, by contrast, often assigns lower confidence to `stable` labels, blurring the distinction between progression categories. With inversion-aware scoring, this gap widens: TILA consistently yields more reliable and well-separated stability scores. Clinically, this capability is crucial for minimizing unnecessary interventions and ensuring robust longitudinal patient management.

## 4 DISCUSSION AND CONCLUSION

Modeling temporal progression in paired chest X-rays remains challenging. We introduced **TILA**, which integrates temporal inversion across pretraining, fine-tuning, and inference. By enforcing order-awareness, TILA improves temporal reasoning with consistent gains across retrieval, zero-shot, few-shot, and supervised tasks. Inversion-aware scoring further stabilizes predictions, and TILA's modular design allows straightforward integration with existing VLP pipelines.

**Limitations.** Label variability is a persistent issue, as radiologists may disagree on subtle changes (Schilling et al., 2022; Nichyporuk et al., 2022; Kang et al., 2022), and many datasets use auto-extracted rather than validated annotations (Irvin et al., 2019; Smit et al., 2020; Jain et al., 2021). Incorporating curated consensus labels would further strengthen reliability.

**On reversed/combined evaluations.** Radiologists assess progression only in the forward direction. Our *Reversed* and *Combined* settings serve as analytical tools to rigorously validate the reliability and trustworthiness of standard predictions.

**Pairwise scope.** TILA currently models adjacent exam pairs, reflecting clinical reporting practice. This design captures clinically meaningful dynamics, but extending to longer sequences using only radiology reports risks label noise, since reports are inherently pairwise. Richer longitudinal modeling may require incorporating metadata such as ICD codes or treatment histories.

**Asymmetry of recovery.** While most findings are reversible, simply reversing a "worsened" case does not guarantee an identical "recovery," as residual abnormalities may persist. Radiology focuses primarily on disease burden, and our results suggest that modeling temporal change explicitly outweighs such asymmetry. Experiments confirm that inversion-aware supervision improves temporal reasoning without harming recognition of recovery.

**Future work.** The principles of TILA extend naturally to CT and MRI. Addressing label variability through consensus or semi-supervised methods and coupling inversion-aware embeddings with multi-modal large language models (Chen et al., 2024; Bannur et al., 2024; Fan et al., 2025) are promising directions toward richer temporal reasoning and more reliable longitudinal patient management.

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

# A APPENDIX

# B MODEL

This section outlines implementation, augmentation, hyperparameters, and model selection for comparison.

## B.1 IMPLEMENTATION DETAILS

For pretraining, we use the AdamW optimizer with a cosine learning rate schedule and a 100-step warm-up. The learning rate is $1 \times 10^{-4}$, batch size 144, and all parameters are in bfloat16. Training was performed on three NVIDIA A6000 GPUs for 30 epochs (totaling 54 GPU-hours).

Fine-tuning also uses AdamW and cosine scheduling, with a 5% warm-up. For linear probing, we set the learning rate to $1 \times 10^{-3}$ and batch size to 32; for full fine-tuning, $1 \times 10^{-5}$ and 128, respectively. Both are run on a single A6000 GPU. Projection layers for both image and text encoders have dimension 128, and the BERT text encoder uses a maximum token length of 256.

## B.2 AUGMENTATION

Image augmentations follow the BioViL-T Bannur et al. (2023) protocol and are applied consistently across pretraining and fine-tuning.

## B.3 HYPERPARAMETERS

Logit scales $\tau$ and $\tau^{swap}$ are initialized to 10, with bias $-10$. We use $W = 1$ for pretraining loss and $\lambda = 50$ for fine-tuning, aligning scales between the BiCE and TCL losses. Change-aware sigmoid loss is enabled after 10 pretraining epochs, and TCL is applied after 20 epochs of fine-tuning to avoid early convergence to the 'stable' class.

## B.4 COMPARED MODELS

We compare only to models with publicly available code, enabling consistent training and evaluation. The CNN+Transformer baseline follows BioViL-T Bannur et al. (2023), with ImageNet pretraining. Methods with insufficient accuracy or unavailable code are omitted.

Comparison models for temporal CXR interpretation remain limited. While many baselines exist for single-image CXR tasks, relatively few address interval change despite its clinical importance. This gap is compounded by the lack of standardized benchmarks, undefined dataset splits, and scarce public code, all of which make reproducible evaluation difficult. These challenges are community-wide rather than unique to our study. To ensure fairness and avoid data leakage, we included only models we could reproduce and evaluate end-to-end under our inversion-aware protocol. Where possible, we also incorporated models with released pretrained weights, even if full training code was not available. Notably, recent papers also compare against a narrow set of baselines, often reusing reported results without independent reproduction. Because our goal is to assess prediction reliability—not just accuracy—and to strictly avoid MS-CXR-T overlap, faithful reproduction was essential.

# C DATA

This section describes dataset splits, label generation protocols, and ethical considerations, including specific details on the **MS-CXR-T_retrieval** benchmark.

## C.1 DATASET SPLITS

We first exclude chest X-ray images without available prior images. The sample counts for each split are presented in table 3. For CheXpert, since official test and validation splits lack corresponding reports or prior images, we use the training set as an external validation set to evaluate retrieval

performance. To minimize sampling bias, we sample 3,000 image pairs from the CheXpert test set, repeating this process 10 times to report the mean and 95% confidence interval. For our private dataset, we collected CXR pairs from 2010–2020, filtering for reports and images containing temporal keywords (e.g., "improved", "worsened", "stable"). Labeling was performed by a researcher with four years of experience in chest X-ray interpretation, and these labels were used for external validation during fine-tuning.

|          | Train   | Validation | Test    |
|----------|---------|------------|---------|
| MIMIC    | 183,302 | 1,330      | 2,871   |
| CheXpert | -       | -          | 123,374 |
| Private  | -       | -          | 2,233   |

Table 3: Data Distribution for MIMIC, CheXpert, and Private

## C.2 DATASET APPROVALS AND ETHICS

For MIMIC data, all LLM-assisted label extraction and the construction of MS-CXR-T$_{\text{retrieval}}$ were performed in strict compliance with PhysioNet guidelines for responsible LLM usage (`https://physionet.org/news/post/gpt-responsible-use`). The private dataset was collected under IRB approval, with all researchers formally registered and authorized for data access.

## C.3 CHANGE LABEL GENERATION

Change/no-change labels were generated using Gemini 2.0 Flash for high accuracy; however, this is a relatively straightforward task that could also be performed by smaller LLMs. Our prompt assigns a label of 1 when a change is detected. For compatibility with the change-aware sigmoid loss, we invert the label ($1{\rightarrow}0$, $0{\rightarrow}1$) before use. Pairs labeled as "-1" are excluded from the training set. The prompt used for label generation is provided below:

> You are given two chest X-ray (CXR) reports: a **previous** CXR report and a **follow-up** CXR report. Your task is to analyze both reports and determine if there are any changes in the follow-up CXR compared to the previous one.
>
> Instructions: 1. Compare the findings in both reports. 2. If there are **any new, worsening, or improving conditions**, return '1'. 3. If the reports state **"no interval change"** or findings are **stable**, return '0'. 4. If unsure, return '-1'. 5. Ensure the output is strictly '0', '1', '-1' without additional text.
>
> Input Format: - **Previous:** [Insert previous report text] - **Follow-up:** [Insert follow-up report text]
>
> Output Format: Return only: - '0' if no changes are detected. - '1' if any changes (new, improved, or worsened findings) are detected. - '-1' if uncertain, or any of the previous report or followup report is not a chest X-ray report.
>
> Few-shot Examples:
>
> **Example 1: No Changes (Output: 0)** - **Previous:** "Left pleural effusion is noted. The cardiac silhouette is normal. No acute abnormalities." - **Follow-up:** "No interval change." - **Output:** '0'
>
> **Example 2: New Finding (Output: 1)** - **Previous:** "The lungs are clear. No pleural effusion or pneumothorax. No focal consolidation. The cardiac silhouette is normal. No acute abnormalities." - **Follow-up:** "A new left lower lobe consolidation is noted, concerning for pneumonia. No pleural effusion or pneumothorax. The cardiac silhouette is normal." - **Output:** '1'
>
> **Example 3: Improvement in Findings (Output: 1)** - **Previous:** "Patchy bilateral infiltrates consistent with pneumonia. No pleural effusion. The heart size is within normal limits." - **Follow-up:** "Bilateral infiltrates have significantly improved. No pleural effusion. The heart size remains within normal limits. - **Output:** '1'
>
> **Example 4: Stable Findings (Output: 0)** - **Previous:** "Mild left basilar atelectasis. No pneumothorax or pleural effusion. No acute cardiopulmonary abnormalities." - **Follow-up:** "Mild left basilar atelectasis remains unchanged. No pneumothorax or pleural effusion." - **Output:** '0'

## C.4 MS-CXR-T$_{\text{RETRIEVAL}}$

We describe here the construction process of **MS-CXR-T$_{\text{retrieval}}$**, our benchmark for evaluating temporal reasoning in CXR report retrieval. The creation pipeline consists of three main stages:

**Stage 1: Splitting Reports.** We first extract the findings section and the corresponding report for each image. Reports are split so that each sentence describes a single radiological finding. This reduces noise in later stages when manipulating progression labels. The prompt used for this sentence-level splitting is as follows:

> Analyze the given radiology report text and split it into sentences, each describing a single radiological finding or view position information. Include both positive and negative findings, as well as view position details, but exclude other non-finding information. Follow these guidelines:

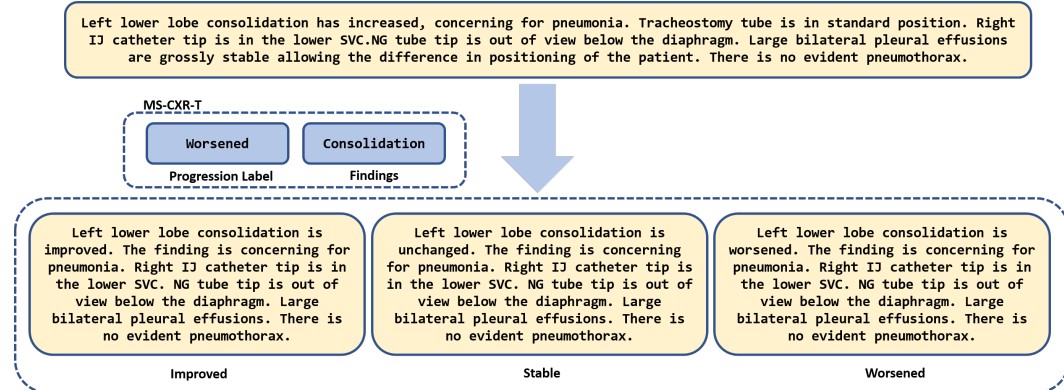

Figure 3: Example workflow for constructing MS-CXR-T$_{\text{retrieval}}$ from the original benchmark.

1. Each sentence should contain only one of the following: (exclude the sentence with view position information) a) A clear radiological finding b) The absence of a specific condition (negative finding)

2. Treat each negative finding (absence of a condition) as a separate observation and split it into its own sentence.

3. Keep sentences that describe view position information, but separate them from findings if they appear in the same original sentence.

4. Exclude sentences that do not describe actual radiological findings or view positions, such as: - Procedural details - General comments about image quality - Patient positioning information (unless it's specifically about the view position)

5. Maintain the meaning and context of the original findings and view positions while splitting.

6. Minor sentence structure changes or addition of necessary words are allowed to ensure clarity.

7. Remove any redundant information and express each finding or view position concisely.

8. Each split sentence should be understandable independently.

9. Avoid using lists or enumerations within a single sentence; instead, create separate sentences for each item.

Example of splitting, including view position, and excluding non-findings: Original Report: '1. A single frontal view of the chest is provided. 2. No consolidation, pleural effusion, or pneumothorax is observed in both lungs. 3. The heart size is normal.' Output: 'No consolidation is observed in both lungs. No pleural effusion is observed in both lungs. No pneumothorax is observed in both lungs. The heart size is normal.'

Process the input text according to these guidelines and return the relevant radiological findings and view position information and do not attach any additional text except the split sentences.

**Stage 2: Prior Reference Omission.** We remove all references to prior studies from each finding. This neutralizes progression status for unrelated findings and allows our benchmark to focus evaluation on the target finding(s) defined in MS-CXR-T. This omission step also enables evaluation in reversed or swapped settings. The prompt used for removing prior references is:

You are an expert chest X-ray (CXR) radiologist familiar with radiologic reports. Your task is to rewrite the given radiology reports by removing all references to prior reports or comparisons, while preserving the original structure as much as possible. Input: A radiology report for a chest X-ray (CXR). Output: A revised CXR report focusing solely on current medical findings, excluding references to prior reports, comparisons, and irrelevant details. Guidelines: Remove Comparisons: Eliminate any terms or phrases that suggest a comparison, such as "compared to," "in comparison with," "change", "cleared", "constant", "decrease", "elevate", "expand", "improve", "decrease", "increase", "persistent", "reduce", "remove", "resolve", "stable", "worse", "new", etc. Focus on Current Findings: Ensure the report only describes the current state of the patient's lungs and related structures. Preserve Medical Context: Maintain the original medical terminology and descriptions of abnormalities. Retain Negations: Keep any negative statements about the absence of abnormalities.

Example: Original Report: The left apex has not been included on this radiograph. The ET tube terminates 3.9 cm above the carina. The NG tube terminates in the stomach. Surgical clips and a faint metallic coil project over the chest. A left PICC terminates in the mid SVC. EKG leads overlie the chest wall. The lung volumes are low. There are persistent bilateral mid and lower zone hazy opacities. There are persistent bilateral hilar and perihilar linear opacities. No significant interval change is observed in the lung opacities. Bilateral pleural effusions are present. The right pleural effusion is greater than the left. No pneumothorax is observed on the right. No cardiomegaly is present. No interval change is observed in the mediastinal silhouette. No significant interval change is observed in the bony thorax. Output: The left apex has not been included on this radiograph. The ET tube terminates 3.9 cm above the carina. The NG tube terminates in the stomach. Surgical clips and a faint metallic coil project over the chest. A left PICC terminates in the mid SVC. EKG leads overlie the chest wall. The lung volumes are low. There are persistent bilateral mid and lower zone hazy opacities. There are bilateral hilar and perihilar linear opacities. Bilateral pleural effusions are present. The right pleural effusion is greater than the left. No pneumothorax is observed on the right. No cardiomegaly is present.

**Stage 3: Creating Progression-Specific Reports.** For each predefined finding of interest, we generate three reports—one each for "improved," "stable," and "worsened" progression. If a specific finding does not exist in the original report, we synthesize the corresponding sentence. The prompt used to generate these progression-specific sentences is:

**Role:** You are an expert assistant specialized in processing medical text, specifically Chest X-Ray (CXR) reports.

**Task:** Given a CXR report text and a specific clinical 'finding', perform the following steps:

1. **Preprocessing:** * Review the input CXR report. * Remove any sentences that *solely* describe the view position (e.g., "PA and lateral views were obtained.", "AP portable view.", "Single frontal view provided."). Do *not* remove view information if it's integrated into a sentence describing a finding (though this is less common). * Retain all sentences describing clinical observations, findings, comparisons, and impressions. Avoid removing other general "unnecessary" words; focus primarily on removing dedicated view position sentences.

2. **Modification based on Finding:** * Identify if the preprocessed report text contains mentions of the provided 'finding'. * **If the finding is mentioned:** * Locate the primary sentence(s) describing the status or appearance of the 'finding'. * Create three versions of the preprocessed report: * **Improved:** Modify the relevant sentence(s) minimally to indicate the finding has 'improved', 'decreased', 'resolved', or similar positive change. * **Stable:** Modify the relevant sentence(s) minimally to indicate the finding is 'stable', 'unchanged', or 'similar'. If the original text already implies stability, ensure this version reflects that clearly. * **Worsened:** Modify the relevant sentence(s) minimally to indicate the finding has 'worsened', 'increased', become 'more severe', or similar negative change. * Make *only the minimal changes* necessary to the specific part about the finding's status. Keep the rest of the report text identical to the preprocessed version. * **If the finding is NOT mentioned:** * Create three versions of the report by appending a new, concise sentence to the end of the preprocessed report text: * **Improved:** Append a sentence like: "The [finding] shows improvement." or "[Finding] is improved." * **Stable:** Append a sentence like: "The [finding] appears stable." or "[Finding] is stable." * **Worsened:** Append a sentence like: "The [finding] has worsened." or "There is worsening of the [finding]." or "[Finding] has increased." * Use the original preprocessed report text for the beginning of each version.

3. **Output:** * Format the final output as a single JSON object string. * The JSON object must have exactly three keys: '"improved"', '"stable"', and '"worsened"'. * The value for each key should be the full text of the corresponding modified report generated in Step 2. Ensure the output is valid JSON.

**Input Format Reminder:** The user will provide input in the following format: 'finding': [The specific clinical finding] 'report': [The full text of the CXR report]

**Example 1 (Illustrative - do not repeat in output):** 'finding': pleural effusion 'report': "When compared to the prior study, the left-sided pleural effusion appears stable. Left consolidation appears relatively stable. No pneumothoraces are seen. The rest of the support lines and tubes are unchanged in position. PA and lateral views were obtained."

**Expected Output Example 1 (Illustrative - do not repeat in output):** '"json { "improved": "When compared to the prior study, the left-sided pleural effusion is improved. Left consolidation appears relatively stable. No pneumothoraces are seen. The rest of the support lines and tubes are unchanged in position.", "stable": "When compared to the prior study, the left-sided pleural effusion appears stable. Left consolidation appears relatively stable. No pneumothoraces are seen. The rest of the support lines and tubes are unchanged in position.", "worsened": "When compared to the prior study, the left-sided pleural effusion is worsened. Left consolidation appears relatively stable. No pneumothoraces are seen. The rest of the support lines and tubes are unchanged in position." }

**Example 2 (Illustrative - do not repeat in output):** 'finding': pneumothorax 'report': "Interval improved aeration is noted at both lung bases. Residual patchy and linear left lower lobe atelectasis remains. A small left pleural effusion is present. Single frontal view."

**Expected Output Example 2 (Illustrative - do not repeat in output):** '"json { "improved": "Interval improved aeration is noted at both lung bases. Residual patchy and linear left lower lobe atelectasis remains. A small left pleural effusion is present. The Pneumothorax is improved.", "stable": "Interval improved aeration is noted at both lung bases. Residual patchy and linear left lower lobe atelectasis remains. A small left pleural effusion is present. The Pneumothorax is stable.", "worsened": "Interval improved aeration is noted at both lung bases. Residual patchy and linear left lower lobe atelectasis remains. A small left pleural effusion is present. The Pneumothorax has worsened." }

Now process the following input:

# D EXPERIMENT

## D.1 ZERO-SHOT PROMPT DESIGN

For each finding and progression class, we design 12–17 distinct prompts to capture the diverse phrasing typically found in radiology reports. Using multiple prompts per class helps reduce score variance, as relying on a single template can lead to unstable results. During zero-shot classification, we compute the cosine similarity between the image representation and each prompt corresponding to a specific progression label, and average these scores to obtain the final prediction.

```
all_prompt={
    'pneumothorax':
        {'improving':["Improved right pneumothorax.",
            "Decreased size of pneumothorax compared to prior.",
            "Interval improvement in pneumothorax.",
            "Partial resolution of left pneumothorax.",
            "Pneumothorax has decreased in size.",
            "Improvement in previously noted pneumothorax.",
            "Marked reduction in size of pneumothorax.",
            "Smaller right apical pneumothorax noted today.",
            "Improved pneumothorax, no acute findings.",
            "Reduction in pneumothorax volume.",
            "Improved appearance of left apical pneumothorax.",
            "Pneumothorax is resolving.",
            "Pneumothorax shows interval decrease."],
```

```
864
865          'stable' : [
866              "Stable small right pneumothorax.",
867              "No increase in size of pneumothorax.",
868              "Pneumothorax appears unchanged from prior.",
869              "Small left pneumothorax, no acute findings.",
870              "No evidence of expanding pneumothorax.",
871              "Pneumothorax is stable with no tension physiology.",
872              "Minimal pneumothorax, no intervention needed.",
873              "Persistent small pneumothorax without progression.",
874              "Pneumothorax noted, patient remains stable clinically.",
875              "No signs of worsening pneumothorax.",
876              "Pneumothorax is stable in appearance and size.",
877              "No interval change in pneumothorax.",
878              "Left apical pneumothorax stable compared to prior.",
879              "Pneumothorax remains small and non-tension."],
880          'worsening' : [
881              "Worsening right pneumothorax.",
882              "Increased size of pneumothorax compared to prior.",
883              "Interval increase in pneumothorax.",
884              "Progression of left pneumothorax.",
885              "Pneumothorax has enlarged.",
886              "Worsening left apical pneumothorax.",
887              "Marked increase in pneumothorax size.",
888              "Pneumothorax increasing, consider intervention.",
889              "Expansion of previously noted pneumothorax.",
890              "Pneumothorax now involves greater lung volume.",
891              "New increase in size of right pneumothorax.",
892              "Pneumothorax shows interval worsening.",
893              "Pneumothorax progressing compared to previous imaging.",
894              "Enlarging pneumothorax noted on follow-up."]
895          },
896      'pleural_effusion':{
897          "improving": [
898              "Improved right pleural effusion.",
899              "Decreased size of pleural effusion compared to prior.",
900              "Interval improvement in pleural effusion.",
901              "Partial resolution of left pleural effusion.",
902              "Reduction in pleural effusion volume.",
903              "Pleural effusion has decreased in size.",
904              "Marked reduction in right pleural effusion.",
905              "Pleural effusion is resolving.",
906              "Improved appearance of left pleural effusion.",
907              "Improvement in previously noted pleural effusion.",
908              "Less fluid seen in pleural space than before.",
909              "Pleural effusion shows interval decrease.",
910              "Decreased right basilar pleural effusion.",
911          ],
912          "stable": [
913              "Stable small right pleural effusion.",
914              "No increase in size of pleural effusion.",
915              "Pleural effusion appears unchanged from prior.",
916              "Small left pleural effusion, no acute findings.",
917              "No evidence of expanding pleural effusion.",
             "Pleural effusion is stable in appearance and size.",
             "Minimal pleural effusion, no intervention needed.",
             "Persistent small pleural effusion without progression.",
             "Pleural effusion noted, patient remains stable clinically.",
             "No signs of worsening pleural effusion.",
             "No interval change in pleural effusion.",
             "Left basilar pleural effusion stable compared to yesterday.",
             "Pleural effusion remains small and unchanged.",
             "Stable bilateral pleural effusions."
         ],
         "worsening": [
             "Worsening right pleural effusion.",
             "Increased size of pleural effusion compared to prior.",
             "Interval increase in pleural effusion.",
             "Progression of left pleural effusion.",
             "Pleural effusion has enlarged.",
             "Worsening left basilar pleural effusion.",
             "Marked increase in pleural effusion size.",
             "Expansion of previously noted pleural effusion.",
             "Pleural effusion increasing, consider intervention.",
             "New increase in size of pleural effusion.",
             "Pleural effusion now causes greater lung compression.",
             "Pleural effusion shows interval worsening.",
             "Pleural fluid accumulation appears progressive.",
             "Enlarging pleural effusion noted on follow-up."
         ]
         },
     'consolidation':{
```

```
        "improving": [
            "Improved right lower lobe consolidation.",
            "Decreased area of consolidation.",
            "Interval improvement in consolidation.",
            "Consolidation has partially resolved.",
            "Marked reduction in pulmonary consolidation.",
            "Clearing of previously noted consolidation.",
            "Consolidation is less extensive than prior.",
            "Improved left basilar consolidation.",
            "Reduction in airspace consolidation.",
            "Consolidation resolving on follow-up imaging.",
            "Less dense consolidation compared to prior.",
            "Airspace opacity improving.",
            "Consolidation has diminished since prior study.",
            "Patchy consolidation appears improved.",
            "Fading consolidation with treatment."
        ],
        "stable": [
            "Stable consolidation in right lower lobe.",
            "No significant change in consolidation.",
            "Persistent left basilar consolidation.",
            "Consolidation appears unchanged from prior.",
            "Airspace opacity remains stable.",
            "No interval change in consolidation.",
            "Chronic consolidation with no acute findings.",
            "Stable patchy consolidation.",
            "Consolidation noted without progression.",
            "No new consolidation identified.",
            "Findings consistent with stable consolidation.",
            "Consolidation remains unchanged.",
            "Stable appearance of parenchymal consolidation.",
            "Consolidation is chronic and stable.",
            "No worsening of consolidation."
        ],
        "worsening": [
            "Worsening right upper lobe consolidation.",
            "Increased area of consolidation.",
            "Consolidation more extensive than prior.",
            "Interval progression of consolidation.",
            "New or expanding consolidation noted.",
            "Consolidation has worsened.",
            "Marked increase in pulmonary consolidation.",
            "Confluent consolidation involving multiple lobes.",
            "Airspace consolidation increasing.",
            "Progressive dense consolidation.",
            "Patchy consolidation more pronounced.",
            "Worsening consolidation despite treatment.",
            "New left basilar consolidation with progression.",
            "Expanding area of alveolar consolidation.",
            "Increased opacification consistent with worsening consolidation."
        ]
    },
    'edema': {
        "improving": [
            "Improved pulmonary edema.",
            "Decreased pulmonary vascular congestion.",
            "Edema appears less prominent than prior.",
            "Interval improvement in pulmonary edema.",
            "Partial resolution of interstitial edema.",
            "Reduction in alveolar edema.",
            "Pulmonary edema has decreased in extent.",
            "Marked reduction in pulmonary edema.",
            "Clearing of previously seen pulmonary edema.",
            "Improved vascular congestion.",
            "Improved interstitial markings.",
            "Pulmonary edema resolving with treatment.",
            "Decreased perihilar opacities.",
            "Edema improving compared to previous study.",
            "Less pulmonary edema seen on current film."
        ],
        "stable": [
            "Stable pulmonary edema.",
            "No significant change in pulmonary edema.",
            "Pulmonary edema appears unchanged from prior.",
            "Edema remains stable in extent.",
            "Persistent mild pulmonary edema.",
            "Pulmonary vascular congestion unchanged.",
            "Interstitial markings stable.",
            "No interval change in pulmonary edema.",
            "Pulmonary edema noted without progression.",
            "No worsening of pulmonary edema.",
```

```
972             "Edema appears chronic and stable.",
973             "Stable vascular congestion.",
974             "No new signs of fluid overload.",
975             "Pulmonary edema similar to previous exam.",
976             "Mild pulmonary edema, no acute change."
977         ],
978         "worsening": [
979             "Worsening pulmonary edema.",
980             "Increased pulmonary vascular congestion.",
981             "Edema appears more prominent than prior.",
982             "Interval increase in pulmonary edema.",
983             "Progressive alveolar edema.",
984             "Pulmonary edema has worsened.",
985             "Marked increase in pulmonary edema.",
986             "Expansion of interstitial edema.",
987             "New or worsening bilateral pulmonary edema.",
988             "Pulmonary edema now more confluent.",
989             "Increasing perihilar opacities.",
990             "Worsening interstitial markings.",
991             "Pulmonary congestion progressing.",
992             "Increased fluid overload signs on imaging.",
993             "Diffuse worsening of pulmonary edema pattern."
994         ]},
995     'pneumonia':{
996         "improving": [
997             "Improved right lower lobe pneumonia.",
998             "Decreased consolidation in left lung.",
999             "Pneumonia shows interval improvement.",
1000            "Clearing of previously seen infiltrates.",
1001            "Reduction in airspace opacity.",
1002            "Partial resolution of pneumonia.",
1003            "Consolidation is less extensive than prior.",
1004            "Improved left basilar pneumonia.",
1005            "Decreased right middle lobe opacities.",
1006            "Pneumonia improving with antibiotic therapy.",
1007            "Pulmonary infiltrates have diminished.",
1008            "Improved patchy opacities.",
1009            "Interval decrease in parenchymal opacities.",
1010            "Airspace disease appears less prominent.",
1011            "Pneumonia resolving compared to previous imaging."
1012        ],
1013        "stable": [
1014            "Stable right lower lobe pneumonia.",
1015            "Pneumonia appears unchanged from prior.",
1016            "Persistent left basilar consolidation.",
1017            "No interval change in airspace disease.",
1018            "Patchy infiltrates stable in appearance.",
1019            "Consolidation remains without significant change.",
1020            "No progression of pneumonia.",
1021            "Airspace opacity unchanged.",
1022            "Stable pneumonia on follow-up imaging.",
1023            "No new consolidation identified.",
1024            "Chronic-appearing infiltrates, no acute change.",
1025            "Stable bilateral patchy opacities.",
                "No worsening of pneumonia noted.",
                "Findings consistent with prior pneumonia, stable.",
                "Stable airspace disease, no interval change."
            ],
            "worsening": [
                "Worsening right upper lobe pneumonia.",
                "Increased consolidation in left lower lobe.",
                "Pneumonia appears more extensive than prior.",
                "Interval progression of pneumonia.",
                "New or worsening bilateral infiltrates.",
                "Airspace opacities have increased.",
                "Expansion of previously seen pneumonia.",
                "Pneumonia worsening despite treatment.",
                "More confluent consolidation noted today.",
                "Increasing parenchymal opacities.",
                "Marked progression of airspace disease.",
                "Increased patchy opacities compared to prior.",
                "Pulmonary infiltrates have progressed.",
                "Worsening left lower lobe pneumonia.",
                "New consolidation suggestive of worsening pneumonia."
            ]
            }
    }
```

## D.2 Additional Experiment on Score Analysis

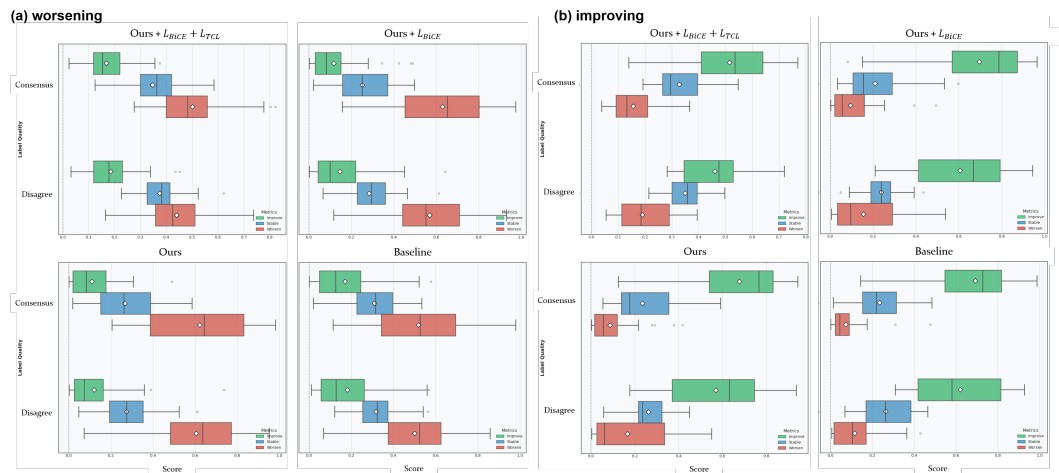

Figure 4: **Finetuning Score Distribution Analysis for Pleural Effusion (MS-CXR-T).** Distribution of model scores for pleural effusion, grouped by label quality ("multiple experts" vs. "disagreement"). Box plots show score distributions for cases labeled as 'worsened' and 'improved', comparing TILA-based models to the baseline. Colors indicate model confidence for each class: green (improved), blue (stable), and red (worsened).

To better understand the impact of temporal inversion and label quality on model outputs, we analyze score distributions from finetuning process for pleural effusion cases in MS-CXR-T. Label quality is categorized as "multiple experts" (consensus among radiologists) or "disagreement" (variation in radiologist assessment). Because disease progression is ordinal, a "worsened" label with disagreement is often closer to "stable".

As shown in fig. 4, models trained with Temporal Consistency Loss (TCL) show a higher overlap of scores in cases with label disagreement, mirroring the inherent ambiguity of expert annotations. In contrast, cases with high consensus result in more confident and well-separated model predictions. These results demonstrate that TILA-based models are both better calibrated and more sensitive to clinical uncertainty, aligning more closely with radiologist interpretation.

## D.3 Clinical Utility of Reversed and Combined Evaluations

Radiologists typically assess temporal progression only in the forward (standard) direction. The *Reversed* and *Combined* evaluations are therefore introduced as analytical tools to rigorously validate the reliability of forward predictions.

The *Reversed* evaluation tests whether a model truly understands temporal progression. For instance, a model reaching 70% accuracy in the forward direction but only 40% in the reversed direction likely exploits spurious cues rather than genuine temporal reasoning. Such discrepancies may undermine the trustworthiness of forward predictions.

The *Combined* evaluation complements this by enforcing consistency across both directions. By aggregating results from forward and reversed pairs, it exposes and helps mitigate directional biases. Together, these settings provide crucial analytical validation, ensuring that predictions in standard clinical scenarios are both reliable and interpretable.

## D.4 Longitudinal Study Scope

Our current formulation focuses on two-time-point image pairs, consistent with radiology practice where the current exam is compared to the immediate prior (*"in comparison to the exam of {date}"*). This reflects guidelines from the American College of Radiology and prior work.

While TILA explicitly models adjacent pairs, it can naturally extend to longer sequences by applying inversion-aware training sequentially across pairs and aggregating embeddings. Several practical applications are possible:

- **Aggregated decision-making:** embeddings from consecutive pairs can form temporal profiles for tasks such as ICD code assignment or diagnosis.
- **Treatment response tracking:** segmenting data around treatment events enables explicit monitoring of therapy effectiveness.
- **Integration with MLLMs:** feeding pairwise embeddings into multimodal large language models could support tasks like narrative generation, treatment adjustment, or outcome forecasting.

These scenarios highlight TILA's flexibility, but extending longitudinal modeling using only radiology reports is difficult, as reports inherently emphasize pairwise comparisons. Applying them to longer sequences risks label noise or hallucination due to limited context. Thus, TILA's main role is to provide accurate, reliable change detection between adjacent exams, establishing a solid foundation for longitudinal tracking. Deeper modeling may require metadata such as ICD codes or treatment records. Similar to BioViL-T, TempA-VLP, SDPL, and CheXRelNet, we use the term *longitudinal* in this restricted sense.

# E ADDITIONAL INFORMATION

## E.1 CODE AND DATASET AVAILABILITY

The dataset will be made publicly available on PhysioNet following approval. All code necessary to reproduce our experiments will also be released upon approval.

## E.2 USE OF LARGE LANGUAGE MODELS

In accordance with ICLR 2026 guidelines, we disclose the use of large language models (LLMs) in this work. LLMs were used in three ways: (i) to refine the clarity and presentation of writing; (ii) to assist in generating change/no-change labels from radiology reports (see Appendix C.3); and (iii) to construct the MS-CXR-T$_{\text{retrieval}}$ benchmark by modifying radiology reports under controlled prompts (see Appendix C.4). All LLM usage was limited to these supporting roles and did not alter the core experimental results or conclusions.

