# OpenReview forum: "Understanding Interval Change in Chest Radiographs via Temporal Inversion"
_ICLR.cc/2026/Conference — ICLR 2026 Conference Withdrawn Submission_

### Official Review · Reviewer_hu3Z · 2025-10-26

**Soundness:** 1
**Presentation:** 2
**Contribution:** 2
**Rating:** 2
**Confidence:** 5

**Summary:**

This paper introduces TILA, a framework designed to improve temporal reasoning in longitudinal chest X-ray analysis. It consistently incorporates temporal inversion across pretraining, fine-tuning, and inference. The paper also proposes a new benchmark, MS-CXR-T retrieval, for evaluating progression-aware retrieval under temporal inversion. Extensive experiments show that TILA achieves good performance across multiple settings, including temporal embedding matching, zero-shot, few-shot, and fully supervised progression classification.

**Strengths:**

The paper presents TILA (Temporal Inversion-aware Learning and Alignment), a novel and well-motivated framework with a new loss function that incorporates temporal inversion across pretraining, fine-tuning, and inference stages for longitudinal chest X-ray analysis. It also introduces a new benchmark, MS-CXR-T retrieval, to evaluate the model’s consistency under temporal inversion. Extensive experiments across multiple datasets and settings demonstrate the robust performance of TILA.

**Weaknesses:**

- In Section 2.1, Equation (1,2) does not clearly define “B, i, and j”, making the loss formulation hard to follow.
- The hyperparameters W=1 and \lambda=50 are fixed without ablation or justification.
- At line 100, the Baseline is described as using the SigLIP loss on BioViL-T; however, in Section 3.4, Table 1, most retrieval metrics of the Baseline are higher than those of TILA, indicating that the claimed effectiveness of the TILA pipeline is insufficient.
- At line 372, the statement that TILA “increases retrieval accuracy” is inconsistent with the results reported in Table 1.
- The paper lacks a clear Ethics Statement, especially regarding the manually annotated tertiary hospital dataset, although some related comments appear at line 721.
- The references to the appendix are inaccurate. Appendix A is cited (line 256) but contains no content, and the appendix guidance is inconsistent with the main text.

**Questions:**

- Could the authors clarify the definitions of “B, i, and j” in Equations (1，2)?
- What is the rationale for choosing the hyperparameters W=1 and \lambda=50? Was any ablation or sensitivity analysis performed to validate these settings?
- Why does the TILA model underperform the Baseline on several retrieval metrics in Table 1, despite the claim that it increases retrieval accuracy?

---

### Official Review · Reviewer_cKAh · 2025-10-27

**Soundness:** 2
**Presentation:** 2
**Contribution:** 2
**Rating:** 2
**Confidence:** 4

**Summary:**

The paper proposes Temporal Inversion aware Learning and alignment to enable temporal awareness by incorporating sensitivity to temporal inversion. The paper further introduces MS-CXR-T retrieval benchmark to measure the reasoning capabilities in retrieval scenarios.

**Strengths:**

The paper tackles an important problem of incorporating the patient history into the multi-modal models. The usage of temporal inversion while lacks clear motivation but is interesting and novel.  The method is simple and shows effectiveness across various evaluation settings. Overall, the paper was easy to read and reasonably written.

**Weaknesses:**

There are multiple aspects of the paper which I believe can be justified better:
* **Pretraining step:** The paper uses two losses (original sigmoid loss and change-aware sigmoid loss) but it is unclear whether both losses are essential. The lack of a) impact of weight $w$ to balance these losses and b) ablation to clarify their requirement makes it hard to justify their need. Further there is a dependency on Gemini 2 Flash but whether the model is reasonable for this task is challenging to quantify and not considered for the overall pipeline.
* **Dependency on the inversion:** It is unclear why inversion is crucial due to various reasons.
     * Consider the case of a patient who is in a critical condition, why is it a necessity for the model to learn the flipped label of improvement when that might not be a possibility for many labels? Why is the reversed direction important for all the labels? Was the clinical relevance of this condition validated? My concern is that this enforcement will encourage scenarios which should not be considered for training the model.
     * How are the cases where the patient is stable/critical but only the intensity of the condition is different can be incorporated in the framework?
* **Comparison with prior works:** The paper lacks a comparison with many recent works that have investigated not only the temporal aspect of radiographs but also reports. For instance, both papers [1,2] conducted an training and evaluation using past images, reports from MIMIC. The utility of enforcing inversion on radiographs is further unclear in cases when such information can directly be obtained from past reports. All the metrics in the paper are biased towards this requirement which makes the positioning of the work unclear. Further, Table 2 is challenging to interpret. My recommendation would be to have individual plots for different findings.

References.
[1] Huang et al., HIST-AID: Leveraging Historical Patient Reports for Enhanced Multi-Modal Automatic Diagnosis. 2024.
[2] Soenksen et al., Integrated multimodal artificial intelligence framework for healthcare applications. 2022

**Questions:**

Please refer to my comments above.

---

### Official Review · Reviewer_6ams · 2025-10-30

**Soundness:** 3
**Presentation:** 3
**Contribution:** 3
**Rating:** 6
**Confidence:** 5

**Summary:**

This paper targeted a significant topic of temporal context in CXR images; this aspect highlights the challenges in longitudinal studies on medical images. The authors proposed the TILA, a framework that can leverage temporal inversions, such as swapping the order of image pairs, as a signal and guidance to enforce disease progression direction. This study used a bidirectional cross-entropy method to predict flip when the image order is reversed. And a TCL, the loss function to enforce the predicted probability distributions is symmetric under inversion.

**Strengths:**

- The authors targeted a well-grounded clinical problem when analyzing CXR images in real practice; the decisions are typically derived from longitudinal series instead of single instances.
- Unlike adopting VLM from the computer vision domain, this paper pre-trained and used a novel temporal inversion as supervision, which can tackle multiple instance CXR at the same time for downstream tasks.
- The results show empirically good performance across zero-shot, few-shot and supervised classification. The author also introduced the TEM score for temporal alignment.
- In general, this paper is well written, methodologically sound.

**Weaknesses:**

- More human-accessible evaluations are needed. LLMs are widely used for benchmarking or labeling, but the temporal disease progression still needs human-standard labels for evaluation.
- Some assumptions are not verified and used for label, such as, the “symmetric progression”, “improved V.S. worsened”. The designed metrics need more explanation.
- CXR is a pioneering modality for VLM and temporal study; there are many existing works on CXR. If the study can go beyond CXR to more radiological modalities, it would be more impactful. The authors can discuss this area.
- Minor: the table, the numbers are too small.

**Questions:**

Questions and suggestions are associated with the weakness section. Thanks.

---

### Official Review · Reviewer_D2KX · 2025-11-01

**Soundness:** 2
**Presentation:** 3
**Contribution:** 3
**Rating:** 6
**Confidence:** 3

**Summary:**

This paper introduces TILA (Temporal Inversion-aware Learning and Alignment), a novel framework designed to enhance temporal reasoning in longitudinal chest radiograph analysis. Unlike prior vision–language pretraining (VLP) models that assess single images in isolation, TILA explicitly models directional disease progression by incorporating temporal inversion—a mechanism that reverses the chronological order of paired chest X-rays during training and inference to enforce logical consistency between “improved” and “worsened” labels. The framework integrates inversion-aware supervision across pretraining, fine-tuning, and inference. Experiments on MIMIC, CheXpert, and a private hospital dataset demonstrate that TILA achieves superior performance across multiple tasks. In addition, the authors constructed a new benchmark, MS-CXR-Tretrieval, specifically designed to evaluate temporal reasoning and progression-aware retrieval in chest radiograph.

Overall, the paper presents a clear and rigorous contribution with some assumptions that merit further validation, but the overall methodology is novel and well-executed.

**Strengths:**

### **1. Comprehensive Pipeline**

Integrates inversion awareness consistently across pretraining, fine-tuning, and inference, forming a coherent end-to-end system rather than an isolated module.

---

### **2. Rigorous Evaluation**

Proposes new evaluation settings (Reversed and Combined) and a dedicated benchmark to assess temporal reasoning—moving beyond simple progression classification accuracy, and includes extensive ablation studies.

---

### **3. Benchmark Construction**

The MS-CXR-Tretrieval benchmark and four evaluation protocols provide a  more rigorous view of model reliability.

---

### **4. Implementation Transparency**

The paper includes detailed implementation information, prompt templates, and dataset construction steps, which  enhance reproducibility and clarity.

**Weaknesses:**

### **1. Simplifying Assumption of Temporal Symmetry**

TILA implicitly assumes that reversing the image order corresponds to a clinically meaningful “opposite” change (e.g., worsening ↔ improvement). The authors do acknowledge the asymmetry of recovery (p.9), but they lack a detailed discussion and quantitative analysis of it.

---

### **2. Limited to Pairwise Modeling**

TILA focuses on adjacent pairs only; it does not yet handle longer temporal chains (e.g., t₀ → t₁ → t₂), which might limit its capacity to model long-term disease trajectories.

---

### **3. No Consideration of Interval Length Between Scans**

The model treats all exam pairs equally, but the time gap (e.g., 1 day vs. 6 months) can drastically change the expected visual and semantic differences, which should be explicitly modeled or analyzed.

**Questions:**

### **1. On the Clinical Symmetry Assumption**

I’m not a radiologist, but from a clinical reasoning perspective, improvement is not always a mirror of worsening (作者). Many pathological findings are **partially irreversible** (e.g., fibrosis, calcification, post-surgical changes), meaning that **reversing the temporal order** of two exams does not necessarily correspond to a true “recovery” scenario.

I fully understand that **modeling temporal order or sequential information between scans** is important and authors do acknowledge the asymmetry of recovery, but I would appreciate if the authors could further justify or elaborate on this:

> **Question:**
> How does *TILA* handle asymmetric progression cases in which the “reverse pair” (T2, T1) is not semantically equivalent to improvement? Could this **symmetry assumption** oversimplify the semantics of disease evolution?

---

### **2. Attribution of Gains**

Since **bidirectional losses** effectively double the number of training passes, the observed improvements might result from **increased optimization stability** rather than inversion-aware supervision itself.

> **Question:**
> Could the authors include a **control experiment** (e.g., forward-only training for doubled epochs) to **disentangle these effects**?

---

### **3. On Temporal Interval (Time Gap) Sensitivity**

It would be helpful if the authors could **report the distribution of time intervals** between paired exams in the datasets (e.g., median, range).

The length of the interval may substantially affect both the **visual appearance** and the **clinical interpretation** of “change”:
- Two scans taken within one day may show only minor variations.
- Scans several months apart may reflect major disease progression or resolution.

> **Questions:**
> - Could the authors comment on how such **interval variability** might influence *TILA*’s performance?
> - If possible, an **analysis of results stratified by time gap** (short vs. long intervals) would make the findings more interpretable and clinically grounded.

---

### Note · Authors · 2025-11-12

**Comment:**

We sincerely thank the reviewers for their thoughtful and constructive feedback. Prior to receiving the reviews, we had already identified several areas for improvement that closely align with the reviewers’ insights. In response, we have undertaken substantial revisions, including more comprehensive comparative analyses, detailed ablation studies, and clarifications to avoid potential overstatements. We have also strengthened the clinical justification for our approach and refined the discussion of TILA’s temporal inversion mechanism to better reflect its practical motivation.

While many of the reviewers’ concerns could be addressed through a standard revision, several updates involve significant methodological and conceptual extensions beyond the scope of the current version. Given the magnitude of these changes, we believe it is more appropriate to withdraw the present submission and resubmit a substantially revised paper that fully reflects these improvements.

We apologize for any inconvenience and sincerely thank the reviewers for their valuable time and insightful feedback, which have greatly contributed to improving the quality and rigor of our work.

**Withdrawal Confirmation:**

I have read and agree with the venue's withdrawal policy on behalf of myself and my co-authors.